# A Systematic Equalizer Design Technique Using Backward Directional Design

**Gihyeon Ji** 

Department of Electrical and Computer Engineering, Hanyang University, Ansan 426-791, Korea;
jkhan2524@hanyang.ac.kr

**Abstract:** This paper presents a systematic equalizer design methodology using a backward directional design (BDD). The proposed design method includes pre-emphasis and crosstalk cancellation design and offers a proper waveform solution for transmitters (TX). Since it is driven by a user-defined specification, it avoids over/under design, reducing wasted power. Furthermore, the proposed design procedure is summarized in systematic algorithms and provides an automated design environment. The procedure has been tested for various line conditions to verify the algorithms. The result shows that the proposed method successfully designs equalizers to within a 2.4% error.

**Keywords:** equalizer design; low power; pre-emphasis; crosstalk cancellation; transmission line

---

## 1. Introduction

The data transfer rate of integrated circuits is increasing [1]. In these systems, signal loss and coupling effects can deteriorate a channel's performance. The high-frequency energy loss and coupling in channels can be reduced by employing pre-emphasis [2–6], decision feedback equalizer [6,7], and crosstalk cancellation (XTC) [8–10], but there remains a problem of designing these circuits. Since there are many different kinds of interconnection in chip systems, a variety of equalizers must be designed to fit their application. This is very complex work, requiring engineers to check specification, power, noise, and area. In particular, the power consumption is the most important factor for duration time, heat generation, and even performance, so it must be designed carefully. Thus, a computer-aided design technique is needed.

A forward directional design (FDD) method is usually used for equalizers. In the FDD process, after the TX circuit is designed, the output response is evaluated. The FDD process may lead to inefficient design iterations or an over-specified design (e.g., in power dissipation and die area) since a suitable input waveform cannot be determined in the early design phase. The design iterations may cause an increase in design cost and time. In contrast, the backward directional design (BDD) process determines the desired output waveform first and thus it minimizes energy waste, implementing an equalizer with low power consumption. However, there is only one existing method, and more research is necessary. The authors of [11] proposed a BDD method for a single data link using only a few poles for the system response calculation, but their method may inaccurately represent high-frequency effects. Further, a new technique is needed to incorporate multi-coupled channel design.

In this paper, a systematic equalizer design methodology based on a BDD method is proposed. It provides more feasible waveforms and more efficient design procedures than previous approaches.

## 2. Conventional Backward Directional Design Method

In a long data transmission channel, a high-frequency signal is distorted due to its low pass nature. A linear system's input and output waveforms can be represented as follows

$$H(\omega, l) = \frac{1}{\cosh(\gamma l) + Z_C/Z_L \times \sinh(\gamma l)}, \tag{1}$$

$$V_{out}(\omega) = V_{in}(\omega) \times H(\omega, l), \tag{2}$$

$$v_{in}(t) = \mathfrak{I}^{-1}\left\{V_{out}(\omega) \times H(\omega, l)^{-1}\right\}. \tag{3}$$

where $H(\omega, l)$ is the transfer function of the interconnect line [12]. $Z_C$ and $Z_L$ are the characteristic impedance and load impedance, respectively. $\gamma$ is the propagation constant. The authors of [11] approximated these expressions using only a few poles within the $K > M$ condition

$$H(s) \approx 1/\sum_{n=0}^{M} a_n s^n, \tag{4}$$

$$V_{out}(s) \approx 1/\sum_{n=0}^{K} b_n s^n. \tag{5}$$

where $a$ and $b$ are coefficients of the transfer function and the output signal, respectively, and $M$ and $K$ are maximum order number. Although this process removes impulse components that cannot be implemented with real circuits, it cannot completely reflect the characteristics of the channel. In particular, high-speed signals cannot be accurately expressed with only a limited number of poles. In addition, a practical output waveform of dense interconnects is closer to an exponential wave shape than it is to waveforms like Figure 1b. Thus, as shown in Figure 1c, the $v_{in}$ waveform is not feasible and exceeds the supply voltage range (0.8 V). To solve these problems, the input and output waveforms have to be designed to consider the characteristics of channels and circuits.

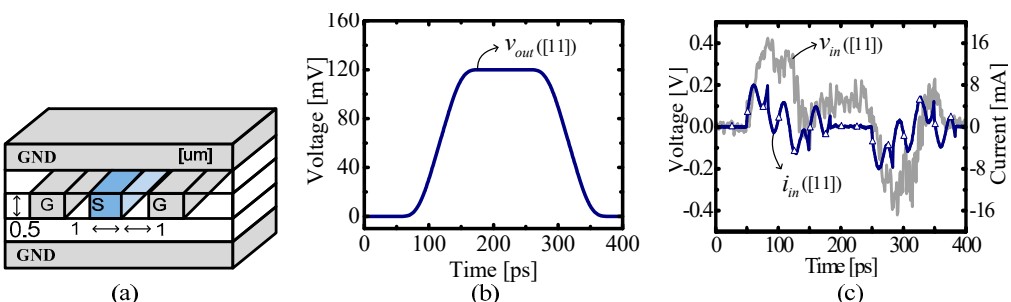

(a)     (b)     (c)

**Figure 1.** Conventional backward directional design (BDD) waveform results. (**a**) Line cross-section. The parameters ($R$ = 34 Ω/mm, $L$ = 0.17 nH/mm, and $C$ = 0.26 pF/mm) are extracted using a SPICE (Simulation Program with Integrated Circuit Emphasis) field solver. (**b**) Output waveforms and (**c**) input waveforms are results of conventional method with (a). Note that the poles of the waveforms are limited by $K$ = 6, $M$ = 5 and that the line length is 6 mm.

## 3. New DBB Design Technique

*Nomenclature*

| | |
|---|---|
| $v_{test}(t)$ | The piece-wise linear test pulse signal. A lowercase letter represents a time-domain signal and an uppercase letter represents a frequency domain signal. |
| $V_{swing}$ | The voltage difference between low and high on the data link. |
| $T_{r(f)}$ | The transient rise (fall) time of the output waveform from 0% to 100%. |
| $l$ | The total line length. |
| $l_k$ | The line length up to the *k-th* line segment $(0 \le k \le m)$ which is given by $l_k = k \cdot \Delta l$, where $\Delta l$ is the length of one line segment. |
| $v_k(t)$ | The waveform of the *k-th* line segment. |
| $v_{desired}(t)$ | The channel output waveform that meets the design criteria. |
| $v_{required}(t)$ | The channel input waveform corresponding to $v_{desired}(t)$. |
| $\widetilde{v}(t)$ | A waveform that applies the relaxation process. |
| $\mathbf{D} = [D_i(\omega)]$ | The n × 1 digital data input matrix of the channel; the matrix size $n$ is the total number of channels. |
| $\{\uparrow 0\}$ | Another expression for $\mathbf{D} = [1\ 0]$. An arrow indicates that a pulse is excited; its direction indicates the sign of a signal; its position indicates channel number (the left is channel 1). |
| $\mathbf{V}_{in} = [V_{in\_i}(\omega)]$ | The $n \times 1$ input waveform matrix of the interconnect. |
| $\mathbf{V}_{out} = [V_{out\_i}(\omega)]$ | The $n \times 1$ output waveform matrix of the interconnect. |
| $\mathbf{H} = [H_{i,j}(\omega)]$ | The $n \times n$ transfer function matrix of the interconnect. |
| $\mathbf{V}_d = [V_{d\_i,j}(\omega)]$ | The $n \times n$ suitable waveform matrix; $V_{d\_i,i}$ indicates a required pre-emphasis waveform for the *i-th* channel; $V_{d\_i,j}$ indicates a required crosstalk cancellation waveform; $\mathbf{V}_{in} = \mathbf{V}_d \cdot \mathbf{D}$. |

---

**Algorithm 1:** Equalizer for Single Line

---

   **Input:** $V_{test}(\omega)$, $H_{line}(\omega,l)$, $T_{r,f}$, $V_{swing}$, $\Delta l$, $f_C$
   **Output:** $\widetilde{V}_{required}(t)$
   **Variables:** $k$, $V_{max}$, $t_{rise,fall}$, $v_k(t)$
  **1:** $k = 0$
  **2:While(True)**
  **3:**    $V_{k+1}(\omega) = V_{test}(\omega) \times H_{line}(\omega,\ (k+1) \times \Delta l)^{-1}$
  **4:**    *Measure $V_{max}$; maximum voltage of $v_{k+1}(t)$*
  **5:**    *Measure $t_{rise}$ and $t_{fall}$ of $v_{k+1}(t)$*
  **6:**   **If** $V_{max} < V_{swing}$
  **7:**    **Break_while**
  **8:**   **End_if**
  **9:**   **If** $t_{ries} < T_r$ & $t_{fall} < T_f$
 **10:**    **Break_while**
 **11:**   **End_if**
 **12:** $k = k + 1$
 **13: End_while**
 **14:** $V_{desired}(\omega) = V_k(\omega)$
 **15:** $\widetilde{V}_{desired}(\omega) = V_{desired}(\omega) \times \{u(\omega) - u(\omega - f_C)\}$
 **16:** $\widetilde{V}_{required}(\omega) = \widetilde{V}_{desired}(\omega) \times H_{line}(\omega,l)^{-1}$

---

### 3.1. Waveform Determination in a Single Line

The first step in a BDD is to define the output waveform. Inappropriate output waveforms may cause inappropriate TX design, so they should be realizable. In this paper, a feasible output waveform is determined from the output response of a practical channel.

In order to find a feasible output waveform, $v_{test}$ is applied to a channel. The test input signal is set to have a transient time of 10% UI (unit interval) and the amplitude $V_{swing}$. Then, every line segment's output waveform is scanned and evaluated against design criteria. During scanning, a

waveform having the largest length while satisfying the criteria is determined to be the desired output waveform. In this paper's method, the design criteria are chosen so that the maximum voltage and the transition time of the output waveform are equal to $V_{swing}$ and $T_{r(f)}$, respectively. Then, a required input waveform $v_{required}(t)$ can be determined simply by substituting $v_{desired}(t)$ into Equation (3). However, this $v_{required}(t)$ does not consider the driver circuit bandwidth and may not be feasible in cases like Figure 1c. This is due to the high-frequency signal components beyond the circuit bandwidth. Therefore, the high frequencies should be removed with a low-pass filter. The filter's cutoff frequency should be lower than the circuit's output bandwidth. This relaxation process has an effect on the edge rate and amplitude of $v_{required}(t)$, so the cutoff frequency should be chosen to make the input waveform feasible. A relaxed input waveform with a lower peak-to-peak value and a lower slew rate is more practical than an unrelaxed waveform that may not be readily implemented. Furthermore, the lower peak-to-peak values and slew rates make output drivers smaller and reduce the power consumption of the equalizer. The foregoing waveform determination procedure is described in Algorithm 1. Note that the proposed technique is a technique for the loss and coupling, and the results may be inaccurate in the case of discontinuity dominant interconnects. Waveforms determined using Algorithm 1 under the conditions of $V_{swing}$ = 120 mV and $T_{r,f}$ = 200 ps are illustrated in Figure 2. The high-frequency components above 12.5 GHz were removed using the relaxation technique.

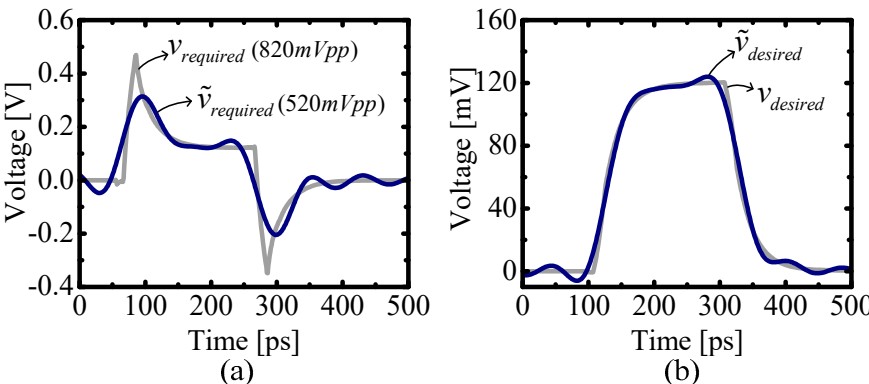

**Figure 2.** Proposed waveform comparison. (**a**) Suitable input waveforms and (**b**) desired output waveforms. The gray line is the unrelaxed waveform and the blue solid line is the relaxed waveform.

### 3.2. Waveform Determination in Multi-Line

In multi-coupled data links, the coupling effect should be taken into account. The crosstalk (XT) noise varies with the gain of the pre-emphasis. In addition, the transmitted pre-emphasis waveform is changed by the added XT cancellation waveform. These unintended variations lead to design uncertainty and make design more difficult. Therefore, these designs require a BDD method. However, a multi-line equalizer design cannot proceed with the same process that is applied to the single-line case. Since a modal decoupling technique cannot be applied to lossy or discontinuous lines [13], it is very difficult to formulate the multi-line transfer function. Thus, this approach uses an unformulated multi-line transfer function consisting of a data set matrix that can be extracted from simulation or measurement. A symmetric multi-line system is schematically described as in Figure 3, and its output waveforms can be represented as

$$\begin{cases} V_{out\_1} = V_{in\_1} \times H_{1,1} + V_{in\_2} \times H_{1,2} \\ V_{out\_2} = V_{in\_1} \times H_{1,2} + V_{in\_2} \times H_{1,1} \end{cases}. \tag{6}$$

In a {↑ 0} data switching pattern, ideal output waveforms are

$$\begin{cases} V_{out1} = V_{desired} \\ V_{out2} = 0 \end{cases}. \tag{7}$$

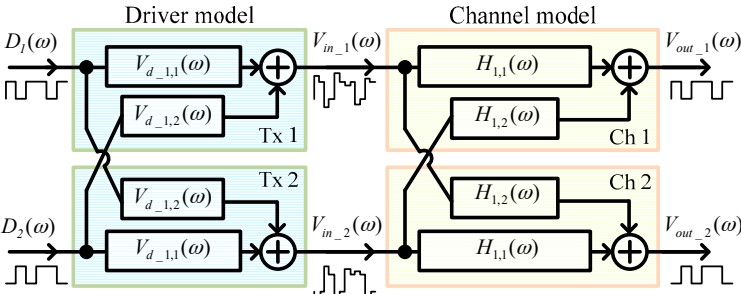

**Figure 3.** Block diagram of signal flow within the symmetric two-coupled line system.

---

**Algorithm 2:** Equalizer for Multi-Lines

---

**Input:** $V_{desired}(\omega)$, **H**
**Output:** $\widetilde{\mathbf{V}}_d$
**Variables:** $k$, $\mathbf{V}_{temp}$
**1:For** $k = 1$ *to 'total numver of the channel'*
**2:**    $\mathbf{V}_{temp} = 0$
**3:**    $V_{temp\_k}(\omega) = \widetilde{V}_{desired}(\omega)$
**4:**    $\left[\widetilde{V}_{d\_k,1}(\omega)\ \widetilde{V}_{d\_k,2}(\omega)\ \widetilde{V}_{d\_k,3}(\omega)\right]^T = \mathbf{H}^{-1} \cdot \mathbf{V}_{temp}$
**5:End_while**

---

So, Equation (6) can be represented as

$$\begin{cases} V_{d\_1,1} \times H_{1,1} + V_{d\_1,2} \times H_{1,2} = V_{desired} \\ V_{d\_1,1} \times H_{1,2} + V_{d\_1,2} \times H_{1,1} = 0 \end{cases}. \tag{8}$$

Then, $V_{d\_1,1}$ and $V_{d\_1,2}$ can be determined as

$$V_{d\_1,1} = V_{desired} \times \frac{H_{1,1}}{H_{1,1}^2 - H_{1,2}^2}, \tag{9}$$

$$V_{d\_1,2} = -V_{desired} \times \frac{H_{1,2}}{H_{1,1}^2 - H_{1,2}^2}. \tag{10}$$

These can be expressed in a closed form as

$$\mathbf{V}_d \cdot \mathbf{D} = \mathbf{H}^{-1} \cdot \mathbf{V}_{out}. \tag{11}$$

Furthermore, Equation (11) can be extended to an *n*-coupled line in the same manner. The details of this process are described as Algorithm 2.

## 4. Verification

### 4.1. Equalizer Design for a Single-Line Data Link

For verification, the test procedure shown in Figure 4 was run with three different line conditions consisting of 3 mm, 6 mm, and 8 mm lengths of the Figure 1a line structure. These conditions represented low, medium, and highly lossy systems. Figure 5 shows the test results. Design criteria were set to $V_{swing}$ = 120 mV and $T_{r,f}$ = 200 ps and the cut-off frequency was 12.5 GHz. The proposed method successfully derived a feasible input waveform. In the SPICE verifications, the error was less than 2.2%. A conventional method [11] was tested with a similar procedure for comparison. The front-end procedure for waveform solution derivation was replaced with the method of [11], and the back-end procedure for SPICE verification was the same. In conventional method's test, $\widetilde{v}_{desired}$ and

$\widetilde{v}_{required}$ were replaced by $v_{out}$ [11] and $v_{in}$ [11], respectively. The conventional waveform results are shown in Figure 6, and a comparison summary is shown in Table 1. In the conventional method, as the loss increased, the error due to the pole approximation increased. In particular, for data over the 6 mm length, an error of more than 26% was occurred, and the feasibility of the waveform was reduced. On the other hand, the proposed method had a consistent accuracy and feasibility over the various lengths. Comparing in the short length (3 mm) data, the proposed method had lower dynamic ranges. This makes the size of the equalizer small, realizing a low power system.

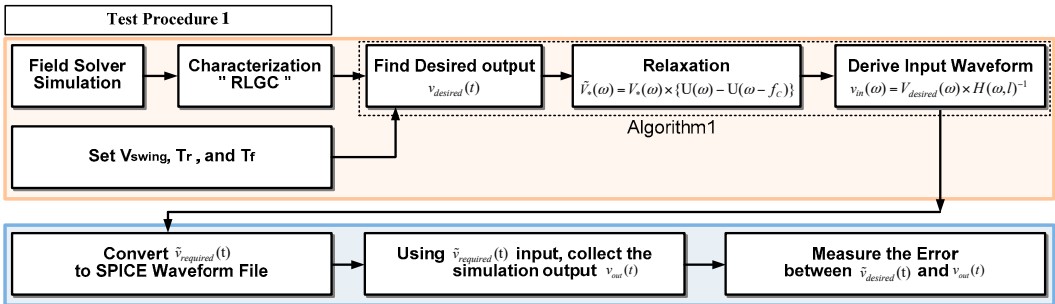

**Figure 4.** Test procedure for a single line.

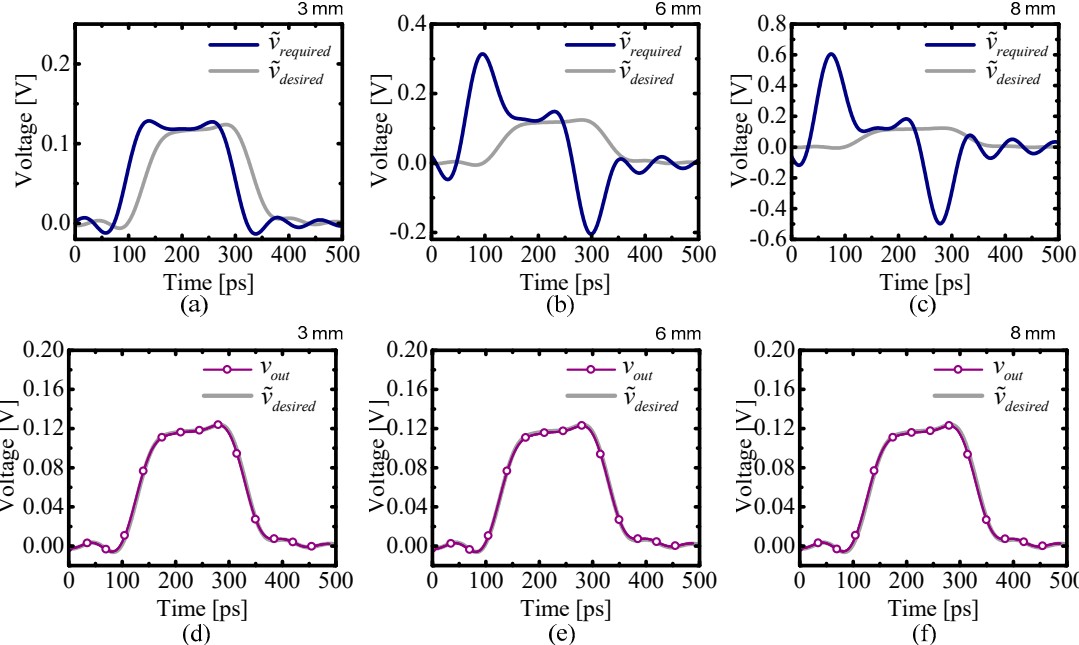

**Figure 5.** Proposed design results. (**a**–**c**) are Algorithm 1 results for 3 mm, 6 mm, and 8 mm length. (**d**–**f**) are SPICE verification results for 3 mm, 6 mm, and 8 mm lengths. Note that the frequency components exceeding 12.5 GHz were relaxed.

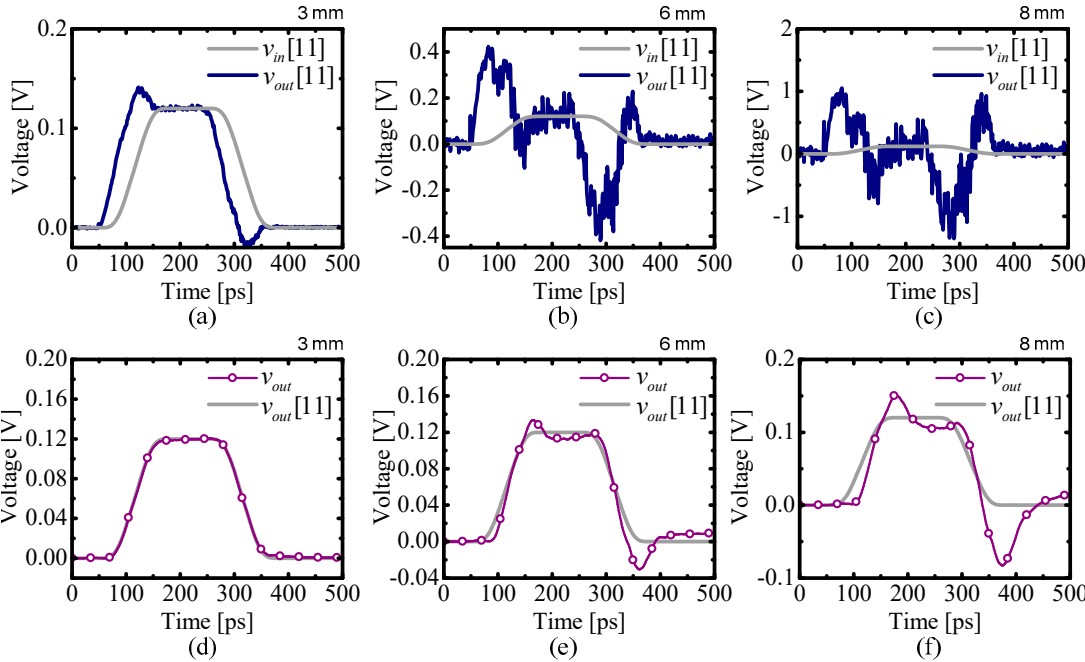

**Figure 6.** Conventional design results. (**a–c**) are waveform design results for 3 mm, 6 mm, and 8 mm lengths. (**d–f**) are SPICE verification results for 3 mm, 6 mm, and 8 mm lengths.

**Table 1.** Comparison summary of single-line design results.

| | Dynamic Range [1] | | | Error [2] | | |
|---|---|---|---|---|---|---|
| | 3 mm | 6 mm | 8 mm | 3 mm | 6 mm | 8 mm |
| Conventional [11] | 0.16 V | 0.84 V | 2.5 V | 2.5% | 26.6% | 69.2% |
| Proposed | 0.14 V | 0.52 V | 1.1 V | 2.1% | 2.2% | 1.9% |

[1] Peak-to-peak voltage of the required input waveform. [2] $Maximun\left(\frac{|\tilde{v}_{desired}(t)-v_{out}(t)|}{V_{swing}} \times 100\right)$.

## 4.2. Equalizer Design for a Multi-Line Data Link

Similar to the verification for a single line, a multi-line test procedure, as seen in Figure 7, was run. The line structure is shown in Figure 8. Since the test line consists of symmetric three-coupled lines, the required input waveforms are shown for only the {↑ 0 0} and {0 ↑ 0} cases in Figure 9. The XTC waveforms corresponding to 2nd order crosstalk noise ($\tilde{v}_{d\_1,3}$ and $\tilde{v}_{d\_3,1}$) are not implemented because they are too small to have a significant effect on the data link system. Using the line parameters and the desired waveform from the single line case, the required input waveforms in the multi-line case and its SPICE verification are shown in Figure 9. The $\tilde{v}_{d\_1,1}$ and $\tilde{v}_{d\_2,2}$ waveforms are used for pre-emphasis; the $\tilde{v}_{d\_2,1}$ and $\tilde{v}_{d\_1,2}$ waveforms are used for XTC. The summary is shown in Table 2. The error is less than 2.4%.

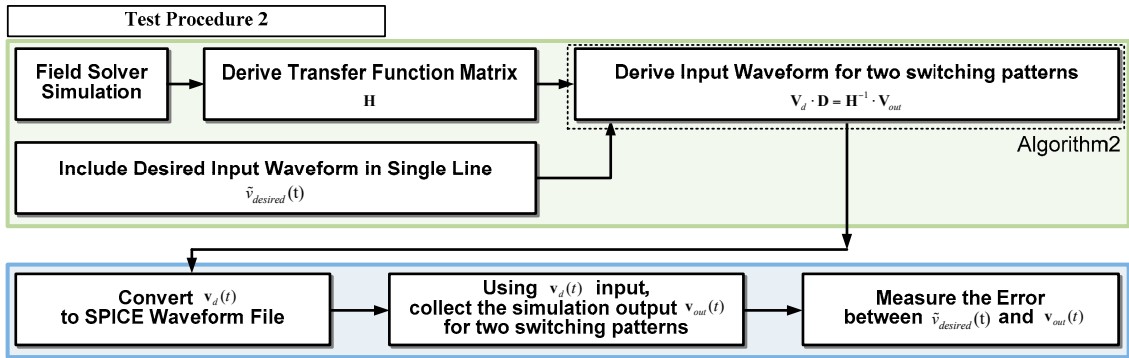

**Figure 7.** Test procedure for a two-coupled line system.

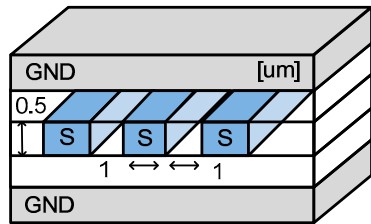

**Figure 8.** Line cross-section of three-coupled lines. The parameters are extracted using SPICE. The resistivity ([$\Omega$/mm]) is $R_{11} = 34$; the inductivities ([nH/mm]) are $L_{11} = 0.16$, $L_{12} = 0.03$, $L_{13} = 0.006$, $L_{21} = 0.03$, $L_{22} = 0.16$, and $L_{23} = 0.03$; the capacitivities ([pF/mm]) are $C_{11} = 0.23$, $C_{12} = 0.06$, $C_{13} = 0.000$, $C_{21} = 0.06$, $C_{22} = 0.19$, and $C_{23} = 0.06$.

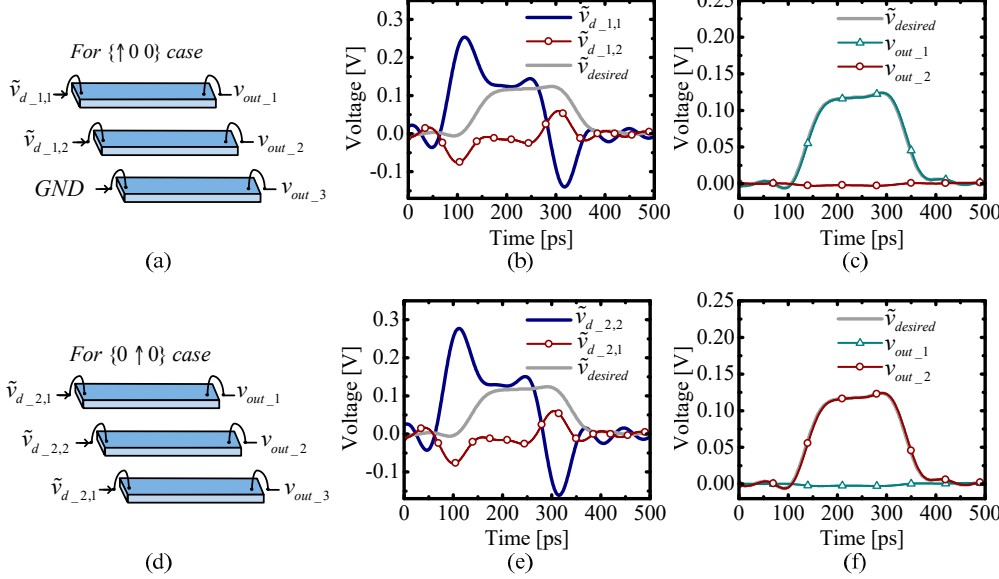

**Figure 9.** Design results for a multi-line equalizer. (**a**,**d**) are test benches for {↑0 0} and {0↑0} cases, respectively. (**b**,**e**) are proposed waveforms using Algorithm 2. (**c**,**f**) are the SPICE verification for each test bench. Note that the frequency components exceeding 12.5 GHz were relaxed.

**Table 2.** Summary of multi-line design results.

| | Dynamic Range [1] | | Error [2] | |
|---|---|---|---|---|
| | Pre-Emphasis | XTC | $v_{out\_1}$ | $v_{out\_2}$ |
| {↑0 0} case | 0.39 V | 0.14 V | 2.1% | 2.4% |
| {0↑0} case | 0.14 V | 0.44 V | 2.4% | 2.2% |

[1] Peak-to-peak voltage of the required input waveform. [2] $Maximun\left(\frac{|\overline{v}_{desired}(t) - v_{out}(t)|}{V_{swing}} \times 100\right)$ for pre-emphasis.

[2] $Maximun\left(\frac{|v_{out}(t)|}{V_{swing}} \times 100\right)$ for XTC.

## 5. Implementation

In this section, further verification is done in the circuit simulation domain. Although this ideal circuit simulation does not represent real circuits, it can show that a proposed waveform can be implemented. Equalizers are implemented to generate the proposed waveforms and verified.

### 5.1. Single Line Equalizer Implementation

Although there are several variants among pre-emphasis circuits, their output waveforms typically have the shape shown in Figure 2a. As an example, the capacitive pre-emphasis [6] was employed for the TX shown in Figure 10, but a feed-forward equalizer circuit could also have been designed by running a least mean squares algorithm for $v_{required}$. The circuit was designed using a 22 nm CMOS process technology library with $V_{DD}$ = 0.8 V [14]. The line cross-section was that of Figure 1a. The data rate and the line length were 5 Gbps and 6 mm, respectively. The required input and the desired output waveforms are shown in Figure 5b. Figure 11 compares the proposed method outputs with circuit simulation outputs. The blue solid line is the SPICE circuit simulation result, whereas the gray line is the MATLAB calculation result using the proposed Algorithm 1. The similarity between the two waveforms indicates that the proposed waveforms are feasible. Eye diagrams are shown in Figure 12.

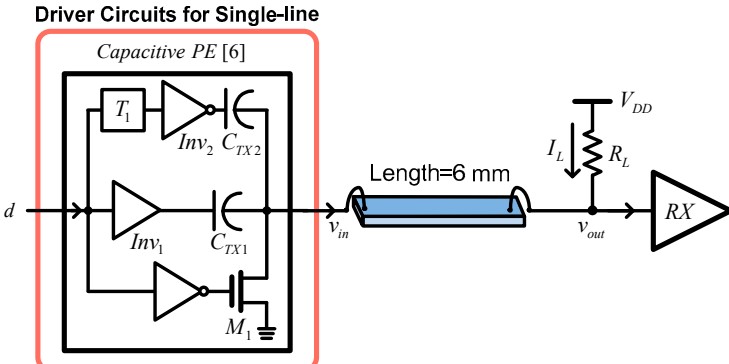

**Figure 10.** Description of the TX for the single-line system. The line is implemented by segment model with resistors, capacitors, and inductors. The detailed circuit parameters are $T_1$ = 74 ps, $Inv_1$ = 10 um/5 um ($Wp/Wm$), $Inv_2$ = 3 um/1.5 um ($Wp/Wm$), $C_{TX1}$ = 350 fF, $C_{TX2}$ = 60 fF, $I_L$ = 20 uA, $R_L$ = 6 kΩ, and $RX$ = 0.3 um $NMOS\ Tr$.

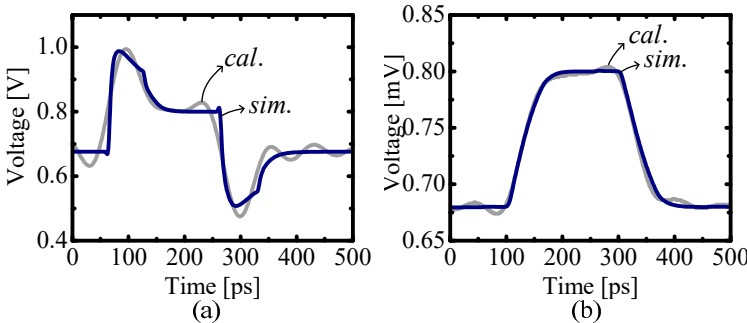

(a)   (b)

**Figure 11.** Comparison of pre-emphasis waveforms for the single-line system. (**a**) Input waveforms and (**b**) output waveforms. The gray lines (cal.) are waveforms determined by the proposed technique, and the blue solid lines (sim.) are simulation results obtained by circuit simulation.

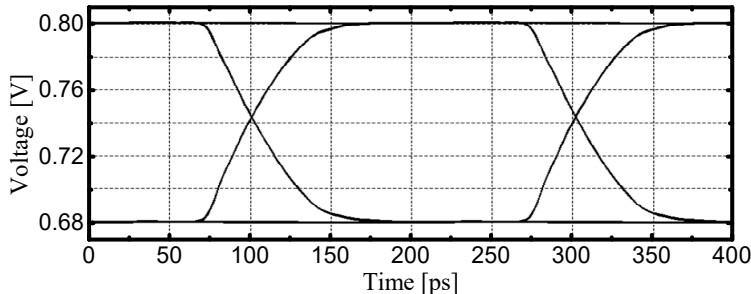

**Figure 12.** Eye-diagrams for a 5 Gbps, 60,000 bits PRBS.

*5.2. Multi-Line Equalizer Implementation*

A multi-line equalizer circuit for three-coupled lines as shown in Figure 13 was tested by simulation as another proposed method. The required input waveforms are shown in Figure 9. The XTC waveforms can be implemented with one of the methods in [8–10]. Ref. [10] was selected this paper. The XTC waveforms were generated by the circuits [10] shown in Figure 13 with the same pre-emphasis circuit as in Figure 10. The proposed waveforms are compared with SPICE simulation in Figure 14, which shows that the proposed waveforms are feasible and can be readily implemented. In addition, the channel loss and coupling noise are successfully compensated. The eye diagrams are shown in Figure 15.

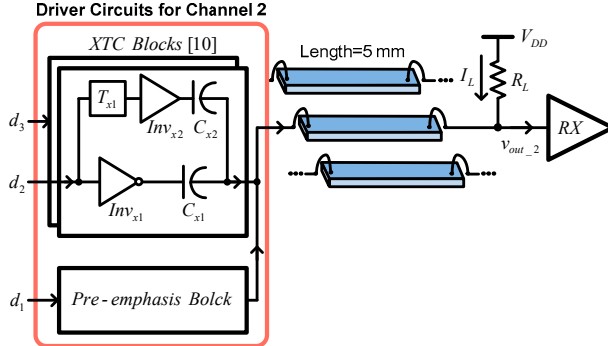

**Figure 13.** Description of the driver circuits for channel 2 and the three-coupled lines. The lines are implemented by RLC segment model. The common circuit parameters are $I_L = 20$ uA, $R_L = 6$ kΩ, and $RX = 0.3$ um *NMOS Tr.*; The pre-emphasis parameters of channel 1 are $T_1 = 85$ ps, $Inv_1 = 12$ um/6 um $(Wp/Wm)$, $Inv_2 = 3$ um/1.5 um $(Wp/Wm)$, $C_{TX1} = 380$ fF, $C_{TX2} = 68$ fF; The XTC parameters of channel 1 are $T_{x1} = 76$ ps, $Inv_{x1} = 8$ um/5 um $(Wp/Wm)$, $Inv_{x2} = 2$ um/1 um $(Wp/Wm)$, $C_{x1} = 100$ fF, $C_{x2} = 50$ fF; The pre-emphasis parameters of channel 2 are $T_1 = 89$ ps, $Inv_1 = 18$ um/9 um $(Wp/Wm)$, $Inv_2 = 3$ um/1.5 um $(Wp/Wm)$, $C_{TX1} = 480$ fF, $C_{TX2} = 115$ fF; The XTC parameters of channel 2 are $T_{x1} = 87$ ps, $Inv_{x1} = 5$ um/2.5 um $(Wp/Wm)$, $Inv_{x2} = 2$ um/1 um $(Wp/Wm)$, $C_{x1} = 100$ fF, $C_{x2} = 50$ fF.

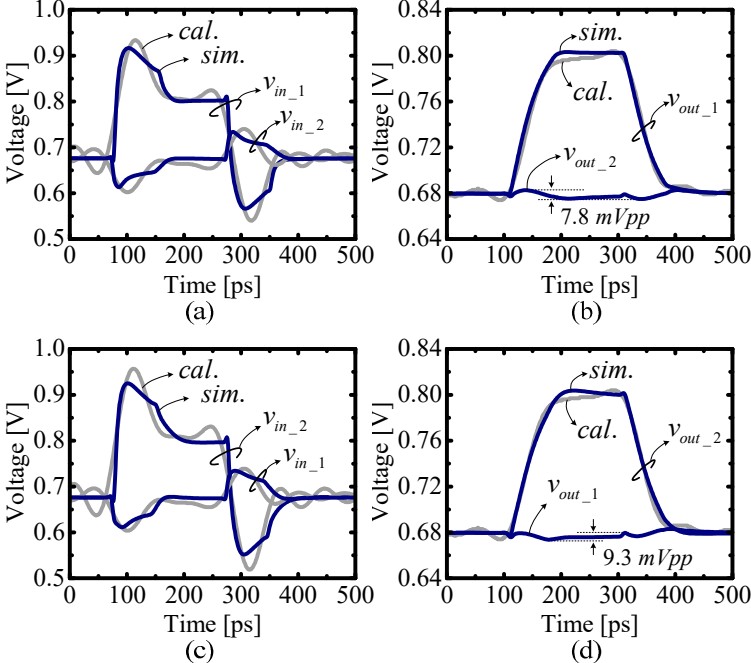

**Figure 14.** Waveforms comparison. (**a**,**b**) are the input and output waveforms for {↑ 0 0} patterns, respectively. (**c**,**d**) are the input and output waveforms for {0 ↑ 0} patterns, respectively. The gray lines (cal.) are waveforms determined by the proposed technique, and the blue solid lines (sim.) are simulation results obtained by circuit simulation.

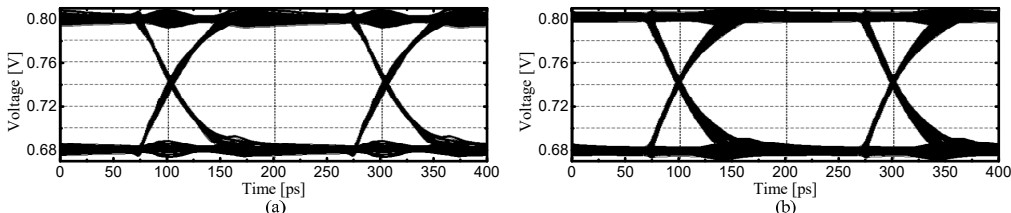

**Figure 15.** Eye diagrams using the proposed technique with a 5 Gbps, 60,000 bit PRBS. (**a**) Channel 1 (outer line) and (**b**) channel 2 (inner line).

## 6. Conclusions

In this paper, a systematic equalizer data link design method was proposed. The method for a single line was presented as Algorithm 1 and used to determine the input waveform to produce the desired output waveform in a practical channel. Then, to achieve a feasible input, the required waveform was re-determined with relaxation. Since relaxed waveforms have a lower dynamic range and edge rate, the implemented equalizer has lower power consumption. With Algorithm 2, the method was extended to multi-line design. Thus, the proposed method can design for pre-emphasis and additionally for crosstalk cancellation. Verification was performed for several line lengths. The results showed that equalizers can be implemented with lower power consumption, and higher accuracy using our design method.

**Author Contributions:** All contributions belong to the author.

**Funding:** This research received no external funding.

**Conflicts of Interest:** The author declares no conflict of interest.

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
