# Peer review of "A Systematic Equalizer Design Technique Using Backward Directional Design"

_electronics, doi:10.3390/electronics8091053_

Round 1

Reviewer 1 Report

A well written and organized paper. Before it can be published, I would suggest authors to add a comparison table to compare your work with similar prior works. This table can provide a clear view on how your work advances the state-of-the-art. Thank you. 

Author Response

Dear reviewer,

Thank you for your sincere comments.

It is my response sheet for you. I hope my answers make you satisfy.

Reviewer 2 Report

Dear author, I can note the high quality of the presented manuscript. I think that your article will be interesting to readers of Electronics journal and very useful for some of them. My comments and suggestions you can find below.

Please, try to avoid emotional expressions in the paper like "dramatically" in line 20. 

Please, format your algorithms on pages 3 and 5 as a code, not as a figure.

The description of the parameters in figures 11, 13 and especially 16 should be moved in the text of the article for better readability.

Have you made comparisons your method with other design methods besides conventional method [11]? Please, write about it in the article.

Also, I would like to read about the limitations and recommendations for the use of your design method. In which cases it is worth using, and in which cases it is worth using the conventional method. Are there conditions when your method will give the worse result? Please, discuss it more.

You write in the conclusion section about faster design time. However, I did not find the results of this estimation in the manuscript. Please, fix it.

Author Response

Dear Reviewer,

Thank you for your sincere comments.

It is my response sheet for you. I hope my answers make you satisfy.
